# Transcriptome Profiling Reveals Differential Expression of Circadian Behavior Genes in Peripheral Blood of Monozygotic Twins Discordant for Parkinson’s Disease

**DOI:** 10.3390/cells11162599

**Published:** 2022-08-20

**Authors:** Ekaterina I. Semenova, Ivan N. Vlasov, Suzanna A. Partevian, Anna V. Rosinskaya, Ivan N. Rybolovlev, Petr A. Slominsky, Maria I. Shadrina, Anelya Kh. Alieva

**Affiliations:** 1Institute of Molecular Genetics of National Research Centre, Kurchatov Institute, 2 Kurchatova Sq., 123182 Moscow, Russia; 2State Public Health Institution Primorsk Regional Clinical Hospital No. 1, 57 Aleutskaya St., 690091 Vladivostok, Russia

**Keywords:** Parkinson’s disease, whole transcriptome analysis, monozygotic twins, gene expression, peripheral blood, circadian rhythms

## Abstract

Parkinson’s disease (PD) is one of the most common neurodegenerative diseases. Investigating individuals with the most identical genetic background is optimal for minimizing the genetic contribution to gene expression. These individuals include monozygotic twins discordant for PD. Monozygotic twins have the same genetic background, age, sex, and often similar environmental conditions. The aim of this study was to carry out a transcriptome analysis of the peripheral blood of three pairs of monozygotic twins discordant for PD. We identified the metabolic process “circadian behavior” as a priority process for further study. Different expression of genes included in the term “circadian behavior” confirms that this process is involved in PD pathogenesis. We found increased expression of three genes associated with circadian behavior, i.e., *PTGDS*, *ADORA2A*, and *MTA1*, in twins with PD. These genes can be considered as potential candidate genes for this disease.

## 1. Introduction

Parkinson’s disease (PD) is one of the most common neurodegenerative diseases [1]. A key feature of PD is the death of dopaminergic (DAergic) neurons in substantia nigra (SN) pars compacta, which leads to an onset of motor symptoms [2]. The clinical diagnosis of PD is made on the presence of classic motor symptoms: tremor, rigidity, bradykinesia, and postural instability [3]. However, it is known that the initiation of PD pathogenesis occurs long before the appearance of the first motor signs [4]. During this period, a range of non-motor symptoms such as hyposmia, sleep disturbances, anxiety, depression, and constipation can manifest. These symptoms are caused by changes in the functioning of various neurotransmitter systems, such as the DAergic, hypocretin, noradrenergic, serotonergic, cholinergic, and histaminergic systems [5,6].

It has been shown that the development of a pathological process in PD may be accompanied by changes in the expression of a number of genes in SN, striatum, and other tissues. One of the most available and perspective material for fundamental and clinical application is peripheral blood. Tyrosine hydroxylase [7], dopamine transporter [8], and dopamine receptors [9] are expressed in blood lymphocytes. The presence of these proteins is characteristic of DAergic neurons. The results of earlier studies with peripheral blood helped to elucidate several steps in PD pathogenesis. On the one hand, these studies confirmed well-known processes involved in PD pathogenesis. These processes include mitochondrial dysfunction, protein ubiquitination [10,11], oxidative stress [10], and apoptosis [12]. Moreover, alteration in lipid metabolism [12,13], membrane transport [14], cytoskeletal function [15], immune response [14,15], iron metabolism [16], regulation of the sleep–wake cycle [17] and some others processes are associated with PD. At the same time, many studies have been carried out to analyze expression changes of individual genes. As a result, a large number of genes that may be involved in the pathogenesis of PD at the expression level were identified [18,19,20].

Investigating individuals with the most identical genetic background is optimal for minimizing the genetic contribution to gene expression during the PD study. These individuals include monozygotic twins discordant for PD. Monozygotic twins have the same genetic background, age, sex, and often similar environmental conditions [21]. To date, a number of studies have been carried out in which molecular markers of PD were studied in discordant monozygotic twins with this disease. Woodard et al. studied one pair of PD discordant twins with a mutation in the *GBA* gene. The authors measured GBA enzyme activity, *MAOB* expression, and dopamine and a-synuclein levels in midbrain DAergic neurons derived from induced pluripotent stem cell (iPSC) [22]. In Kaut et al.’s work, DNA methylation analysis was carried out in the peripheral blood mononuclear cells of twins discordant for PD, among which there were 12 pairs of exactly monozygotic twins [23]. Mazzetti et al. studied the accumulation of a-synuclein oligomers in skin biopsies of 19 pairs of monozygotic twins discordant for PD [24]. Dulovic-Mahlow et al. investigated mitochondrial phenotypes in skin fibroblast cultures from five pairs of monozygotic twins discordant for PD [25]. In addition, in our previous studies, a whole transcriptome analysis was performed in the fibroblasts [26], as well as in the iPSCs and neuronal progenitor cells (NPCs) [27], of monozygotic twins discordant for PD. However, studies on the analysis of transcriptomic profiling of the peripheral blood of such twins have not been conducted yet. In this regard, the aim of the present study was to carry out a transcriptome analysis of the peripheral blood of three pairs of monozygotic twins discordant for PD.

## 2. Materials and Methods

### 2.1. Patients

The study involved three pairs of phenotypically and genetically monozygotic twins discordant for PD. Information about the symptoms of twins with PD are presented in Table 1. More detailed information about patients is presented in the work carried out earlier [26]. All blood samples were collected with the informed consent of the investigated subjects. The study was conducted according to the guidelines of the Declaration of Helsinki and was approved by the Ethics Committee of Institute of Molecular Genetics of the National Research Centre, the “Kurchatov Institute”, protocol code N°3 19 February 2018.

### 2.2. Sequencing and RNA-Seq Data Analysis

Sample preparation for sequencing and RNA-seq data analysis were performed, as previously described [26]. Sequencing was performed using HiSeq1500 (Illumina, San Diego, CA, USA), obtaining no less than 5 mln. 50 bp reads per library. When analyzing RNA-seq data for one of the twins, one high quality sequence was not obtained. In this regard, several libraries of reads were obtained for it, which were collected into one after checking for correlation between repetitions (Spearman’s R^2^ > 0.95). Trimmed FASTQ files were aligned on the transcriptome that was obtained from the GRCH38 genome and GRCH38.91 gene annotation.

### 2.3. RNA Isolation and Expression Analysis of Individual Candidate Genes

Total RNA isolation from peripheral blood was performed, according to the previously described protocol [14]. After isolation, yeast tRNA was added to the resulting solution of total RNA with a concentration of 1 mg/mL [28].

Analysis of mRNA levels using reverse transcription and real-time PCR (TaqMan technology) was carried out, in accordance with the protocols described previously [26]. Real-time PCR was carried out using QuantStudio 3 cycler (Applied Biosystems, Waltham, MA, USA).

### 2.4. Statistical Processing and Bioinformatic Analysis

Gene Ontology Biological Processes (GO BP) [29] term enrichment was carried out using the apps ClueGO v. 2.5.8 [30] and CluePedia v. 1.5.8 [31] for Cytoscape v. 3.9.0. Significantly enriched terms were selected based on one-sided hypergeometric tests with Benjamini-Hochberg correction (*p*-value < 0.01). Term groups were formed based on common genes per term (>50%). GO BP terms from the 3rd to 6th levels of hierarchy with percentage of associated genes > 20% were considered for enrichment. Term clusters were selected based on common genes. Selection of genes associated with PD was conducted using Pathway Studio v. 12.4.0.5 (Elsevier, Amsterdam, The Netherlands). The request included the keyword “Parkinson”.

Primer and probe sequences design was produced using Beacon designer 7.0 software (Premier Biosoft International, Palo Alto, CA, USA) and the nucleotide sequences of the *MTA1*, *TP53*, *ADORA2A*, *NR1D1*, *OPRL1*, *PTGDS*, *AHCY*, *GPR157*, *NAGLU*, and *SRD5A1* genes and the housekeeping genes *SARS* and *PSMD6* from the NCBI database. Checking the specificity of primers and probes was performed using Primer3 and BLAST (https://www.ncbi.nlm.nih.gov/tools/primer-blast/, accessed on 6 July 2022) [32]. The sequences of gene-specific primers and probes are presented in Table 2.

The protocol of statistical analysis is described in detail, in the work carried out earlier [25]. Interaction network of studied genes was built in Pathway Studio v. 12.4.0.5 (Elsevier, Amsterdam, the Netherlands) using key words “Parkinson”, “neurodegeneration”, “mitochondria”, “oxidative stress”, “apoptosis”, “autophagy”, and “protein ubiquitination”.

## 3. Results

In the present study, we performed a transcriptomic profiling of the peripheral blood of three pairs of monozygotic twins discordant for PD. In the first stage of our study, RNA-seq analysis was carried out. We obtained 1512 differentially expressed genes (DEGs). For these DEGs, an enrichment analysis was performed using the GO BP database. Identified biological processes are presented in Table 3.

Table 3 shows that obtained biological processes form nine groups were not connected with each other. For genes from these groups, the possible involvement in the pathogenesis of PD was assessed using the Pathway Studio program. The results of this assessment are presented in the last column of Table 3. It shows that the largest percentage of genes associated with PD is in the “circadian behavior” group. In this connection, this group was chosen by us for further more detailed research.

In the second stage of our study, we analyzed changes in expression at the mRNA level using real-time PCR for genes included in the circadian behavior term: *ADORA2A*, *AHCY*, *GPR157*, *MTA1*, *NAGLU*, *NR1D1*, *OPRL1*, *PTGDS*, *SRD5A1*, and *TP53*. At the same time, it should be noted that the representation of transcripts of the *AHCY*, *GPR157*, *NAGLU*, and *SRD5A1* genes was lower than the detection level of the method used in the work. The results of the expression analysis for other genes are presented in Table 4.

As shown in Table 4, a statistically significant increase in the relative levels of mRNA expression was obtained only for three genes (*ADORA2A*, *MTA1*, and *PTGDS*).

## 4. Discussion

Currently, several studies have been published on the analysis of monozygotic twins discordant for PD. In these works, attention was paid to the study of changes in the levels of dopamine and a-synuclein in the midbrain DAergic neurons derived from the iPSC of twins [22], DNA methylation in the peripheral blood mononuclear cells [23], accumulation of a-synuclein oligomers in skin biopsies [24], and mitochondrial phenotypes in skin fibroblast cultures [25]. In addition, in one of the above-mentioned works, an RNA-seq analysis was performed in monozygotic twins discordant for PD [22]. However, it should be noted that these twins are carriers of a mutation in the *GBA* gene. We are also actively conducting whole-transcriptome studies of different cell types of monozygotic twins discordant for PD and not carrying mutations associated with this disease. To date, data on the study of fibroblasts [26], as well as iPSC and NCP [27], have already been published. In this work, we carried out an RNA-seq analysis of the peripheral blood.

During the whole-transcriptome analysis, we identified a number of processes (Table 3), among which “circadian behavior” was chosen as a priority process for more detailed study. This choice was due to the highest percentage of genes in the group that can be associated with PD. In total, the “circadian behavior” term contained 10 genes. As can be seen from Figure 1, 9 out of 10 genes may be associated with the pathogenesis of PD. At the same time, no direct association with PD has been shown for the *AHCY* gene, but there is evidence of its involvement in processes that are characteristic of this disease (oxidative stress and apoptosis).

Currently, it is known that circadian rhythm dysfunction can be observed in PD [33]. It is described that, in general, patients with PD are characterized by reduced amplitude of the rest–activity cycle [34]. They often have nocturnal hypertension, reversed circadian blood pressure rhythm, and disruptions in circadian thermoregulation and in hormone rhythms [35]. The characteristics of the sleep–wake cycle are the main behavioral markers of circadian rhythms. Sleep–wake disturbances are the most common group of non-motor symptoms in patients with PD. They include insomnia, excessive daytime sleepiness, rapid eye movement sleep behavior disorder, and restless legs syndrome [36]. Furthermore, at the molecular level, a change in the expression of some circadian genes was shown in patients with PD [37,38,39,40,41]. Thus, the change in the expression of genes included in the “circadian behavior” term that we identified may indicate the involvement of this process in PD and confirms the previously obtained data. For further analysis of expression, six genes were taken from the “circadian behavior” term with a level of transcript representation sufficient for detection: *ADORA2A*, *MTA1*, *NR1D1*, *OPRL1*, *PTGDS*, and *TP53*. As shown in Table 4, only three of the studied genes showed a statistically significant increase in the relative levels of mRNA, which indicates their possible role in the pathogenesis of PD.

The highest change in expression was observed for the *PTGDS* gene. This gene encodes prostaglandin D2 synthase, which catalyzes the isomerization of prostaglandin H2 to prostaglandin D2 (PGD2) [42]. At present, it is known that the PTGDS-PGD2-DP1 receptor (DP1R) pathway is involved in the regulation of sleep [43]. It has been shown in model animals that increased expression of *PTGDS* leads to an increase in the phase of non-REM sleep [44], and PGD2 has a sleep-inducing effect [45,46]. Increased levels of PTGDS are observed in patients with narcolepsy and excessive sleepiness [47]. In the present work, we observed a 2.7-fold increase in *PTGDS* expression in twins with PD relative to healthy twins. The significant increase in *PTGDS* expression that we found may indicate possible disorders associated with the duration of different sleep phases in twins with PD and lead to increased sleepiness. In addition, the work on human neuroblastoma cell cultures revealed that PTGDS can have anti-inflammatory and antioxidant functions and prevent cell death caused by excessive accumulation of reactive oxygen species [48,49]. The anti-inflammatory role of PTGDS was also confirmed in a mouse study of brain tissues and astrocyte cultures. In particular, it was noted that the DJ-1 protein associated with PD exerts an anti-inflammatory effect precisely through the regulation of *PTGDS* expression [50]. Therefore, PTGDS can potentially prevent oxidative stress and apoptosis observed in neurodegenerative diseases, in particular in PD. The increase in *PTGDS* expression in our work may be due to the development of compensatory mechanisms against increased oxidative stress and inflammation. It has now been proven that DAergic neurons die in the SN in PD, and processes such as oxidative stress and inflammation are involved in the death of these neurons [51]. Our results correlate with data obtained in previous studies [11,17]. However, it should be noted that the data of these studies were obtained by a whole-transcriptome analysis without followed verification using real-time PCR. Based on the above, it can be assumed that changes in *PTGDS* expression may be important in the pathogenesis of PD.

To date, there is evidence that the PTGDS-PGD2-DP1R and *ADORA2A* pathways, interacting with each other, are involved in the regulation of sleep. In rodent studies, it has been demonstrated that signaling through the A2A adenosine receptor is required for PGD2-induced sleep [52,53]. In addition, administration of PGD2 in mice led to an increase in their extracellular level of adenosine due to stimulation of the DP1R receptor [46]. In the present work, we observed a 1.8-fold increase in *ADORA2A* expression in twins with PD. The protein encoded by the *ADORA2A* gene is a G-protein coupled adenosine receptor of A2A subtype. The A2A receptor is expressed in the brain, predominantly in the striatum, where it influences the functioning of neurons. It has been reported that the injection of an A2A receptor agonist into different areas of the rodent brain significantly increases the length of non-REM and REM sleep. This effect may be mediated by the fact that A2A receptor activation inhibits histaminergic transmission, by increasing GABA release in the tuberomammillary nucleus [54]. The increase in *ADORA2A* expression that we found may affect the sleep–wake cycle in twins with PD. As in the case of the *PTGDS* gene, it is likely that an increase in *ADORA2A* expression will lead to a somnogenic effect. Since in our work we observed the simultaneous increase in *PTGDS* and *ADORA2A*, it can be assumed that the protein pathways of these genes interact with each other in PD as well.

In recent years, the A2A adenosine receptor has been increasingly cited as a potential target for the treatment of PD symptoms [55,56]. In this case, a positive effect is achieved due to the inhibition of A2A. However, the protective mechanism of ADORA2A inhibition in PD remains unclear [57]. One possibility is that a change in A2A receptor signaling is important for α-synuclein-induced activation of astrocytes and NF-kB. An intra-hippocampal injection of mutant α-Syn fibrils in mice caused induction of A2A receptor expression as well as massive gliosis and neuroinflammation [58]. *ADORA2A* knockout attenuated synuclein aggregate formation, astrogliosis, NF-κB activation, and neuronal apoptosis [58,59]. Thus, suppression of the A2A receptor activity will prevent the development of inflammatory processes leading to neuronal death. It has been proven that intracellular accumulation of α-synuclein aggregates is one of the key characteristics of PD [51]. In this regard, the 1.8-fold increase in *ADORA2A* expression in twins with PD may be due to the accumulation of α-synuclein and may contribute to reducing the pathological process. In addition, it is known that the ADORA2A receptor can be involved in motor function due to interaction with the dopamine D2 receptor [60]. It has been shown that ADORA2A receptors can form heterodimers with D2 receptors, thereby suppressing DAergic signaling [61]. Hence, we suggest that an increase in *ADORA2A* gene expression may have a negative impact on motor function in patients with PD. Our results are in agreement with previous studies, where expression level of the A2A receptor increased both at the mRNA level and at the protein level in patients with PD [62,63,64].

*MTA1* was another gene for which a statistically significant change in expression was observed in the present study. *MTA1* expression was 1.5 times higher in PD twins relative to their healthy siblings. The *MTA1* gene encodes metastasis-associated protein 1. MTA1 modulates the expression of target genes by functioning as a corepressor or coactivator [65]. A study in mice found that MTA1 was involved in the regulation of circadian rhythms through the regulation of transcription of the *CRY1* gene. In addition, MTA1 can interact with the CLOCK-BMAL1 heterodimer and recruit it to its promoter containing the E-box element. This mechanism stimulates the transcription of *MTA1* itself. Interestingly, *MTA1* knockout mice lost the rhythmic expression of core circadian genes [66]. As far as circadian rhythm disturbances are often observed in PD, including changes in the expression of core circadian genes, we suggest that increase in *MTA1* expression in twins with PD may be an adaptive mechanism for maintaining circadian rhythm at the molecular level. Furthermore, MTA1 protein acts as an upstream coactivator of tyrosine hydroxylase (*TH*), the main enzyme of dopamine synthesis. MTA1 physically achieves this function by forming an initiation complex together with DJ1 and PolII on the *TH* promoter [67]. In our study, twins with PD showed a 1.5-fold increase in *MTA1* expression relative to healthy twins. An increase in *MTA1* expression can lead to an increase in *TH* expression with a subsequent increase in the intensity of dopamine synthesis. Thus, we suggest that MTA1 may be involved in the development of compensatory mechanisms in PD. To date, only in the work of Kumar et al. was the expression level of *MTA1* in PD analyzed, which showed the decreased expression of *MTA1* in SN in patients with PD [68]. It should be noted that post-mortem brain samples were used in Kumar et al.’s work. Accordingly, it can be assumed that the patients were in the late, most severe stages of PD, when active drug treatment was completed and the presence of comorbid diseases is possible. It is likely that at later stages of the disease, SN cells lose the resources necessary to maintain compensatory mechanisms, which is the reason for the lower level of *MTA1* expression.

## 5. Conclusions

We performed the first transcriptomic profiling of the peripheral blood of monozygotic twins discordant for PD, who were not carrying mutations associated with this disease. During the enrichment analysis, we identified the metabolic process “circadian behavior” as a priority process for further study. Different expression of genes included in the term “circadian behavior” confirms that this process is involved in PD pathogenesis. Three genes from the term “circadian behavior”, *PTGDS*, *ADORA2A*, and *MTA1*, demonstrated a significant increase in expression at the mRNA level in twins with PD. These genes can be considered as potential candidate genes for this disease, and, in future works, it will be necessary to analyze their expression levels in independent samples of patients with PD.

## Figures and Tables

**Figure 1 cells-11-02599-f001:**
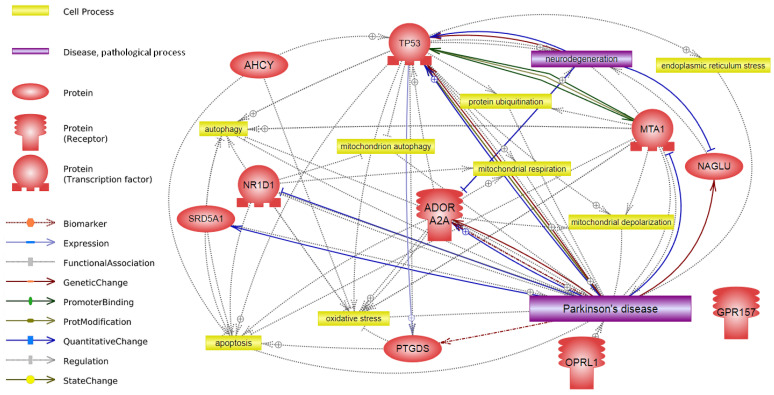
Interaction network of circadian behavior genes. The network was built using Pathway Studio v. 12.4.0.5.

**Table 1 cells-11-02599-t001:** Characteristics of twins with PD.

	Patient No. 1	Patient No. 2	Patient No. 3
Sex	Female	Female	Female
Year of birth	1956	1947	1949
Disease onset date	2002	2007	2011
PD stage (Hoehn–Yahr scale)	2	4	3
Motor symptoms:			
Tremor	Yes	Yes	Yes
Rigidity	Yes	Yes	Yes
Bradykinesia	Yes	Yes	Yes
Postural instability	Yes	Yes	Yes
Mimic disorders	Hypomimia	Hypomimia	Hypomimia
Handwriting changes	Micrographia	-	-
Speech problems	Yes	-	-
Non-motor symptoms:			
Olfactory dysfunction	Hyposmia	-	-
Sleep disturbances	-	Insomnia	Insomnia, nightmares
Emotional disturbances	-	Anxiety, depression	Anxiety
Cognitive deficits	-	Memory impairment, hallucinations	Memory impairment, hallucinations
Pain	Yes	Yes	-
Constipation	-	-	Yes
Urinary dysfunction	-	-	Yes

**Table 2 cells-11-02599-t002:** Sequences of gene-specific primers and probes.

Gene	Nucleotide Sequence
*SARS1*	Probe: 5′-VIC-TCGCCACTCGCTGTCTGCCTTCACCA-BHQ2-3′
(*Seryl-TRNA Synthetase 1*)	Forward primer: 5′-CCCAGCCCTCATCCGAGAG-3′
NM_001330669.1 *	Reverse primer: 5′-TGTTCAAGTTGTCTGCCCGAAATC-3′
*PSMD6*	Probe: 5′-VIC-AGGCGGTTTCTCCTGTCCCAGTCTCCTC-BHQ2-3′
(*Proteasome 26S Subunit*, *Non-ATPase 6*)	Forward primer: 5′-AACACAGAAAAGGCCAAAAGCTTAAT-3′
NM_001271779.1 *	Reverse primer: 5′-AATAGCCACACAATAAAGACCCTGAT-3′
*MTA1*	Probe: 5′-VIC-ATTTCCCCTTCCTCGCCGTTGTCCG-BHQ2-3′
(*Metastasis Associated 1*)	Forward primer: 5′-ACATCTCCAGCACCCTCATCG-3′
NM_004689.4 *	Reverse primer: 5′-TCGGGCAGGTCCACCATTT-3′
*TP53*	Probe: 5′-VIC-GTGTGGTGGTGCCCTATGAGCCG-BHQ2-3′
(*Tumor Protein P53*)	Forward primer: 5′-GCGTGTGGAGTATTTGGATGAC-3′
NM_001126114.2 *	Reverse primer: 5′-ATGTAGTTGTAGTGGATGGTGGTA-3′
*ADORA2A*	Probe: 5′-VIC-AATGATGCCCTTAGCCCTCGTGCCG-BHQ2-3′
(*Adenosine A2a Receptor*)	Forward primer: 5′-CATCGCCATTGACCGCTACA-3′
NM_001278497.1 *	Reverse primer: 5′-GTTCCAACCTAGCATGGGAGTC-3′
*NR1D1*	Probe: 5′-VIC-GTGATGACGCCACCTGTGTTGTTGTTG-BHQ2-3′
(*Nuclear Receptor Subfamily 1 Group D Member 1*)	Forward primer: 5′-CCAGTTTGAATGACCGCTCTCA-3′
NM_021724.5 *	Reverse primer: 5′-GCTGCCATTGGAGTTGTCACTA-3′
*OPRL1*	Probe: 5′-VIC-GCTCCTGGGGAACTGCCTTGTCA-BHQ2-3′
(*Opioid Related Nociceptin Receptor 1*)	Forward primer: 5′-CATCGTGGGGCTCTACCTG-3′
NM_001318853.2 *	Reverse primer: 5′-ATTGGTGGCTGTCTTCATTTTGG-3′
*PTGDS*	Probe: 5′-VIC-TTCACAGAGGATACCATTGTCTTCCTGCC-BHQ2-3′
(*Prostaglandin D2 Synthase*)	Forward primer: 5′-GGAGAAATTCACCGCCTTCTG-3′
NM_000954.6 *	Reverse primer: 5′-AGCCCTGGGGAGTCCTATT-3′
*AHCY*	Probe: 5′-VIC-CATTGTGTGGATGCTGAAACTGAACCC-BHQ2-3′
(*Adenosylhomocysteinase*)	Forward primer: 5′-TAGTTCATCAAGTTGCTACCAGAGT-3′
NM_001322086.2 *	Reverse primer: 5′-TACCGCTCCCGCATACG-3′
*GPR157*	Probe: 5′-VIC-GCCTCGCACAGATCGCCTG-BHQ2-3′
(*G Protein-Coupled Receptor 157*)	Forward primer: 5′-CTCTACTTGTACCTCAGCATCG-3′
NM_024980.5 *	Reverse primer: 5′-GCGTCATAGCCAATCTTCTTCA-3′
*NAGLU*	Probe: 5′-VIC-CGCTCCTTCGGCATGACCCCA-BHQ2-3′
(*N-Acetyl-Alpha-Glucosaminidase*)	Forward primer: 5′-CCCCTCCTGGCACATCAAG-3′
NM_000263.4 *	Reverse primer: 5′-GCCCATCTTCGTGACATTGAC-3′
*SRD5A1*	Probe: 5′-VIC-TTCCTCCTCGCATCAGAAATGGGT-BHQ2-3′
(*Steroid 5 Alpha-Reductase 1*)	Forward primer: 5′-ATGGTCAGAATGGAAACAAATAACAAG-3′
NM_001324323.2 *	Reverse primer: 5′-GCCGTTACAGGTACAGAACATAA-3′

* Accession numbers in the GenBank database (NCBI-GenBank Release 246.0). VIC—fluorescent dye; BHQ2—fluorescence quencher.

**Table 3 cells-11-02599-t003:** The results of enrichment analysis of data obtained by RNA-seq analysis of the peripheral blood of twins discordant for PD.

GO Term (GO ID)	Benjamini-Hochberg Adjustment *p*-Value of Hypergeometric Test for Enrichment	GO Group ^1^	Number Of DEG, Associated with GO Group	Group *p*-Value Corrected with Benjamini-Hochberg	Percentage of Genes in Group, Associated with PD ^2^
circadian behavior (GO:0048512)	7.05 × 10^−3^	I	10	2.38 × 10^−3^	80.00
negative regulation of glucose catabolic process to lactate via pyruvate (GO:1904024)	5.15 × 10^−3^	II	6	2.10 × 10^−3^	50.00
glucose catabolic process to lactate via pyruvate (GO:0019661)	7.82 × 10^−3^				
negative regulation of mitophagy (GO:1901525)	7.82 × 10^−3^				
lactate metabolic process (GO:0006089)	8.56 × 10^−3^				
response to testosterone (GO:0033574)	5.13 × 10^−3^	III	12	4.33 × 10^−3^	41.66
cellular response to testosterone stimulus (GO:0071394)	9.62 × 10^−3^				
protein transmembrane import into intracellular organelle (GO:0044743)	8.25 × 10^−3^	IV	8	3.31 × 10^−3^	25.00
positive regulation of ligase activity (GO:0051351)	8.83 × 10^−3^	V	4	3.31 × 10^−3^	25.00
regulation of triglyceride biosynthetic process (GO:0010866)	7.91 × 10^−3^	VI	6	2.60 × 10^−3^	16.66
regulation of ATP biosynthetic process (GO:2001169)	8.61 × 10^−3^	VII	6	3.33 × 10^−3^	16.66
cytoplasmic sequestering of protein (GO:0051220)	5.19 × 10^−3^	VIII	7	1.95 × 10^−3^	14.29
cytoplasmic sequestering of transcription factor (GO:0042994)	8.56 × 10^−3^				
branched-chain amino acid metabolic process (GO:0009081)	7.39 × 10^−3^	IX	8	2.53 × 10^−3^	0
leucine metabolic process (GO:0006551)	7.46 × 10^−3^				

^1^ Term groups were formed based on common genes per term (>50%). ^2^ Evaluation of the possible involvement of genes from the GO Group in the pathogenesis of PD based on the analysis of the literature data using Pathway Studio.

**Table 4 cells-11-02599-t004:** Relative mRNA levels of studied genes in the peripheral blood of twins discordant for PD.

Genes	Peripheral Blood
*ADORA2A*	**1.81**^1^ **1.34**–**2.16**^2^
*MTA1*	**1.54** **1.05**–**1.86**
*NR1D1*	0.77 0.36–1.2
*OPRL1*	1.19 1.08–2.08
*TP53*	0.91 0.69–1.23
*PTGDS*	**2.67** **2.25**–**2.98**

^1^ median, ^2^ 25–75 percentiles. The data in bold are statistically significant (*p* < 0.05). The expression level in the control is taken as 1.

## Data Availability

Raw and processed data can be accessed in the Gene Expression Omnibus, accession number GSE208347.

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
