# Peer review of "Transcriptome Profiling Reveals Differential Expression of Circadian Behavior Genes in Peripheral Blood of Monozygotic Twins Discordant for Parkinson’s Disease"

_cells, 2022, doi:10.3390/cells11162599_

Round 1
Reviewer 1 Report
Review of a manuscript “Transcriptome Profiling Reveals Differential Expression of Circadian Behavior Genes in Peripheral Blood of Monozygotic Twins Discordant for Parkinson's Disease” by Semenova and coauthors submitted to “Cells”.
Parkinson’s disease is the second prevalent after Alzheimer’s disease neurodegenerative disorder, for which there is still no efficient treatment affecting the course of the illness. The study of the genetic contribution to the pathogenesis of Parkinson’s disease allows to better understand its mechanism and signaling pathways implicated in its progression. The authors performed a transcriptome analysis of peripheral blood of three pairs of monozygotic twins discordant for Parkinson’s disease. This approach allowed identification of the metabolic process «circadian behavior» as a priority process for further investigations. This is an important area of biomedical research and the results presented in the manuscript will be interesting for the readership of “Cells”.
The following corrections and additions should be done.
Abstract
“Three genes from the term «circadian behavior», PTGDS, ADORA2A, and MTA1…” The sense of the sentence is unclear and should be corrected. If the authors want to say “Three genes controlling «circadian behavior», PTGDS, ADORA2A, and MTA1…” they should be say so. Alternatively, the sentence may be corrected as follows “We found increased expression of three genes associated with circadian behavior, i.e., PTGDS, ADORA2A, and MTA1, in twins with PD.
Introduction
-After the sentence “The clinical diagnosis of PD is made on the presence of classic motor symptoms: tremor, rigidity, bradykinesia, and postural instability” the authors should add the following citation:”Biomarkers in Parkinson’s Disease”. Chapter in a book. Peplow P.V., Martinez B., Gennarelli T.A. (eds) Neurodegenerative Diseases Biomarkers. 2022. Neuromethods, vol 173. pp 155-180. Humana, New York, NY. https://link.springer.com/protocol/10.1007/978-1-0716-1712-0_7
-“Earlier studies conducted with peripheral blood proved to clarify pathogenesis of PD.” The sentence should be corrected as follows:” The results of earlier studies with peripheral blood helped to elucidate several steps in PD pathogenesis.”
-The style of the following sentence should be corrected “On the other hand, the association between PD pathogenesis and lipid metabolism [11,12], membrane transport [13], cytoskeletal function [14], immune response [13,14], iron metabolism [15], regulation of the sleep-wake cycle [16] and some others has been shown”. It should be rewritten as follows: ”Moreover, alterations in lipid metabolism [11,12], membrane transport [13], cytoskeletal function [14], immune response [13,14], iron metabolism [15], the sleep-wake cycle [16] and some others processes are associated with PD”.
Figures.
Figure 1 Purple color designated as “Disease” cannot be applied to “neurodegeneration”, this should be corrected.
Discussion
“The significant increase in PTGDS expression that we found may indicate possible disorders associated with the duration of different sleep phases in twins with PD and lead to increased sleepiness.”
It would be interesting to know if the correlation may be found between the biochemical changes described by the authors and real changes in sleep behavior in PD patients studied by the authors.
Overall, this is an interesting manuscript containing new data. The author's team represents professionals in the area of their research.
Author Response
Response to Reviewer 1 Comments
The authors thank the distinguished reviewer for the interest to the work and for the important remarks.
R1 – Point 1.
Abstract. “Three genes from the term «circadian behavior», PTGDS, ADORA2A, and MTA1…” The sense of the sentence is unclear and should be corrected. If the authors want to say “Three genes controlling «circadian behavior», PTGDS, ADORA2A, and MTA1…” they should be say so. Alternatively, the sentence may be corrected as follows “We found increased expression of three genes associated with circadian behavior, i.e., PTGDS, ADORA2A, and MTA1, in twins with PD.
Authors response 1:
Thank you very much for this comment. We have made corresponding correction in the abstract.
R1 – Point 2.
Introduction. -After the sentence “The clinical diagnosis of PD is made on the presence of classic motor symptoms: tremor, rigidity, bradykinesia, and postural instability” the authors should add the following citation:”Biomarkers in Parkinson’s Disease”. Chapter in a book. Peplow P.V., Martinez B., Gennarelli T.A. (eds) Neurodegenerative Diseases Biomarkers. 2022. Neuromethods, vol 173. pp 155-180. Humana, New York, NY. https://link.springer.com/protocol/10.1007/978-1-0716-1712-0_7
Authors response 2:
We added this reference in the introduction.
R1 - Point 3.
-“Earlier studies conducted with peripheral blood proved to clarify pathogenesis of PD.” The sentence should be corrected as follows:” The results of earlier studies with peripheral blood helped to elucidate several steps in PD pathogenesis.”
Authors response 3:
This sentence has been paraphrased as recommended by the reviewer.
R1 - Point 4.
-The style of the following sentence should be corrected “On the other hand, the association between PD pathogenesis and lipid metabolism [11,12], membrane transport [13], cytoskeletal function [14], immune response [13,14], iron metabolism [15], regulation of the sleep-wake cycle [16] and some others has been shown”. It should be rewritten as follows: ”Moreover, alterations in lipid metabolism [11,12], membrane transport [13], cytoskeletal function [14], immune response [13,14], iron metabolism [15], the sleep-wake cycle [16] and some others processes are associated with PD”.
Authors response 4:
Changes have been made in the introduction in accordance with your suggestion.
R1 - Point 5.
Figures. Figure 1 Purple color designated as “Disease” cannot be applied to “neurodegeneration”, this should be corrected.
Authors response 5:
This designation of "neurodegeneration" is automatically set by Pathway Studio, because the program equates this term with a group of diseases. However, we have corrected the wording of this designation in the figure legend to make it the most accurate.
R1 - Point 6.
Discussion.“The significant increase in PTGDS expression that we found may indicate possible disorders associated with the duration of different sleep phases in twins with PD and lead to increased sleepiness.” It would be interesting to know if the correlation may be found between the biochemical changes described by the authors and real changes in sleep behavior in PD patients studied by the authors.
Authors response 6:
There are no data of increased sleepiness and disorders associated with the duration of different sleep phases in the anamnesis of twins with PD. Thus, we cannot say that there is any correlation between the theoretical justification for the increase in PTGDS expression we obtained and actual sleep changes in the patients.
Reviewer 2 Report
Investigating monozygotic twins discordant for PD may significantly help to better understand the underlying pathological processes.
However, there are some questions/comments concerning the manuscript.
Information about the patients should be included the article, either in the Materials and methods section or in the supplement. The characteristics of their disease should be included as well. Do they suffer from motor/non-motor symptoms?
It is mentioned that “at the molecular level, a change in the expression of some circadian genes was shown in patients with PD [36-38].” More recent results/reviews would be also worth mentioning (such as Fifel K, Videnovic A. Front Neurosci. 2021 Jan 18;14:627330. doi: 10.3389/fnins.2020.627330. ).
The authors discussed the possible role of the identified genes in PD. Previously, the carried out transcriptome analysis of dermal fibroblasts and induced pluripotent stem cells ([25,26]) in the cases of the same patients if I understood correctly. Have they found alterations in similar genes or in different ones? The obtained results may be mentioned and compared in the Discussion section as well.
Author Response
Response to Reviewer 2 Comments
The authors thank the distinguished reviewer for the interest to the work and for the important remarks.
R2 – Point 1.
Information about the patients should be included the article, either in the Materials and methods section or in the supplement. The characteristics of their disease should be included as well. Do they suffer from motor/non-motor symptoms?
Authors response 1:
This is a really important addition. Thank you for this idea. Information about patients and their symptoms has been added to the text as a table (materials and methods section).
R2 - Point 2.
It is mentioned that “at the molecular level, a change in the expression of some circadian genes was shown in patients with PD [36-38].” More recent results/reviews would be also worth mentioning (such as Fifel K, Videnovic A. Front Neurosci. 2021 Jan 18;14:627330. doi: 10.3389/fnins.2020.627330.).
Authors response 2:
We added references to more recent studies, including the work suggested by the reviewer.
R2 - Point 3.
The authors discussed the possible role of the identified genes in PD. Previously, the carried out transcriptome analysis of dermal fibroblasts and induced pluripotent stem cells ([25,26]) in the cases of the same patients if I understood correctly. Have they found alterations in similar genes or in different ones? The obtained results may be mentioned and compared in the Discussion section as well.
Authors response 3:
In our previous works, we performed transcriptome analysis in fibroblasts as well as in iPSCs and neuronal progenitor cells of the same twins as in the present study. We, of course, performed comparisons of the results obtained, but did not detect changes in expression for similar genes. In addition, the identified metabolic processes from the enrichment analysis were also different. Since similar results were not obtained in different cell types, we have not reflected this in the discussion text.